# Impact of Communicative and Informative Strategies on Influenza Vaccination Adherence and Absenteeism from Work of Health Care Professionals Working at the University Hospital of Palermo, Italy: A Quasi-Experimental Field Trial on Twelve Influenza Seasons

**DOI:** 10.3390/vaccines8010005

**Published:** 2019-12-24

**Authors:** Claudio Costantino, Alessandra Casuccio, Francesca Caracci, Stefania Bono, Giuseppe Calamusa, Gianmarco Ventura, Carmelo Massimo Maida, Francesco Vitale, Vincenzo Restivo

**Affiliations:** Department of Health Promotion Sciences, Maternal and Infant Care, Internal Medicine and Medical Specialties (PROMISE) “G. D’Alessandro”, University of Palermo, Via del Vespro 133, 90127 Palermo, Italy; alessandra.casuccio@unipa.it (A.C.); fracaracci@hotmail.it (F.C.); stefania.bono01@unipa.it (S.B.); giuseppe.calamusa@unipa.it (G.C.); gianmarco.ventura@unipa.it (G.V.); carmelo.maida@unipa.it (C.M.M.); francesco.vitale@unipa.it (F.V.);

**Keywords:** influenza vaccination, health care workers, vaccination coverage, communication campaigns, informative strategies

## Abstract

Every year, about 20% of health care workers (HCWs) acquire influenza, continuing to work and encouraging virus spreading. Influenza vaccination coverage rates and absenteeism from work among HCWs of the University Hospital (UH) of Palermo were analyzed before and after the implementation of several initiatives in order to increase HCWs’ awareness about influenza vaccination. Vaccines administration within hospital units, dedicated web pages on social media and on the UH of Palermo institutional web site, and mandatory compilation of a dissent form for those HCWs who refused vaccination were carried out during the last four influenza seasons. After the introduction of these strategies, influenza vaccination coverage went up from 5.2% (2014/2015 season) to 37.2% (2018/2019 season) (*p* < 0.001), and mean age of vaccinated HCWs significantly decreased from 48.1 years (95% CI: 45.7–50.5) to 35.9 years (95% CI: 35.0–36.8). A reduction of working days lost due to acute sickness among HCWs of the UH of Palermo was observed. Fear of adverse reactions and not considering themselves as a high-risk group for contracting influenza were the main reasons reported by HCWs that refused vaccination. Strategies undertaken at the UH of Palermo allowed a significant increase in vaccination adherence and a significant reduction of absenteeism from work.

## 1. Introduction

Nosocomial infections due to influenza viruses are a considerable public health problem, especially concerning the most fragile patients (i.e., immunocompromised or hospitalized in Intensive Care Units), associated with a high morbidity and mortality [1].

Patients hospitalized in public structures, or assisted at home, can be infected by other patients, visitors, or health care workers (HCWs) [2,3].

During the influenza season, it has been estimated that about 20% of HCWs contract influenza virus, often continuing to work, leading not only to nosocomial epidemics but also to workforce reduction with consequentdisruptions for the medical care [4,5].

The Public Health authorities have clearly affirmed the importance of vaccination as the most effective measure to prevent the influenza virus spread among the general population [6,7]. Despite International recommendations and recognized efficacy and safety of the vaccine, influenza vaccination coverage rates among European HCWs continues to be less than 75%, the minimum rate able to limit nosocomial transmission of the disease [8,9].

The most important reasons for influenza vaccination refusal were predominantly attributable to poor knowledge, attitudes, and perception of HCWs about vaccination efficacy and safety [10,11,12]. Moreover, gaps in understanding the complex mix of factors leading to vaccine hesitancy among health-care professionals emerged, and it should be contrasted only with tailored strategies in order to increase vaccination awareness [13,14].

According to the WHO strategy for addressing vaccine hesitancy, a multistep approach could be the more effective intervention to increase vaccination coverage rates [15]. For this reason, several studies were conducted on HCWs based on multilevel strategies as communication, health education, and promotion, as well as easy access to vaccination [16,17]. The main limitation of these studies was the short-term evaluation, considering that vaccination confidence needs a long time to show their impact among the general population and, of consequence, also among HCWs.

The present study evaluates the trend of influenza vaccination coverage rates during the last twelve influenza seasons among health care workers attending University Hospital (UH) of Palermo, Italy, estimating the impact of the communicative and informative strategies implemented on vaccination adherence and, as a consequence, on absenteeism from work due to acute sickness.

## 2. Materials and Methods

A quasi-experimental field trial was conducted at the UH of Palermo through the implementation of several interventions to increase vaccination coverage.

From influenza season 2007/2008 to 2018/2019 (from the 1st of November to the 31st of March), influenza vaccination coverage rates and absenteeism from work, among HCWs of the UH of Palermo, were analyzed.

To evaluate absenteeism from work, the following three parameters, due to acute sickness arisen during the influenza season, were seasonally evaluated:Number of HCWs absent from work;Number of overall working days lost;Number of working days lost for single HCWs of the UH of Palermo.

Furthermore, a compulsory anonymous online consent/dissent form was administered, from the middle of October to the end of November, during 2016/2017, 2017/2018, and 2018/2019 influenza seasons. For HCWs that accepted to receive the influenza vaccination, an online reservation system ensured the day and the data for vaccine administration. On the other hand, HCWs that refused vaccination filled in the mandatory dissent form [18].

In particular, the online dissent form was administered, in accordance with the Decree of the Sicilian Health Department, to all health-care professionals working at the UH of Palermo in their personal web page (restricted to unauthorized users), and, in the case of not-completing, a limited access to reserved areas of monthly wages sheets and attendances overview was applied [18]. Response rates among HCWs that declined influenza vaccination were 63.9% (2016/2017 season), 72.6% (2017/2018 season), and 87.5% (2018/2019 season).

HCWs that not responded to the online form and that declined vaccination were signaled to the Regional Health Department at the end of every season. Administrative personnel and all the staff members of the UH of Palermo without any direct contact with patients were excluded from the study.

The dissent form consisted of a single item examining the main reason for influenza vaccination refusal.

The research project, with the informed consent and the dissent form, has been approved by the Ethical Committee of University Hospital of Palermo, Italy, in the month of September 2015 (n.10/2015).

### 2.1. Communicative and Informative Strategies for HCWs Adopted at the UH of Palermo

Before and during the vaccination campaigns, a team of Public health medical doctors, in agreement with the Regional Decree number 1849 of 2016, was dealing with many initiatives to raise knowledge and awareness on influenza vaccination effectiveness and safety among HCWs [18].

In particular, as reported in Table 1, the strategies adopted during the last four seasons were:-Duringthe 2015/2016 season, pins were distributed (with the logo “I’m vaccinated”) to vaccinated HCWs, to hang on the white coat;-During 2016/2017 and 2017/2018 seasons, dedicated days were arranged to influenza vaccination promotion (“flu vaccination day”), endorsed by the Sicilian Health Department in all the Region, with the purpose to spread information on influenza vaccination trough a conference open to HCWs and the general population held at the University Hospital of Palermo;-From the 2016/2017 campaigns to date, posters and flyers that promoted influenza vaccination uptake of HCWs and patients were hungin every hospital unit of the Palermo UH, with the original slogan of the campaign: “Protect yourself to protect your patients”;-During the last four seasons, dedicated pages on social networks (such as Facebook^®^ and Instagram^®^) and on the institutional web site of the UH of Palermo were provided to promote information about influenza vaccination campaign and to invite all health care professionals (including trainees and medical residents) to be vaccinated [19]. Moreover, all vaccinated HCWs were invited to take part in the social media campaign “Show your face”, letting themselves be photographed with a specific “hashtag”, such as #ivaccinemyself, #iamvaccinated, #protectyourpatients, #dontbeinginfluenced. At the end of every immunization day, all the authorized pictures were posted in the dedicated social network pages, and HCWs were asked to share the post in their personal diaries to raisethe visibility of the campaign;-During vaccination seasons 2015/2016, 2016/2017, 2017/2018, and 2018/2019, tailored weeks were also organized to carry out influenza vaccines within hospital units (defined “on-site vaccination”) to make vaccination easier for those professionals that were unable to leave their workplace;-Finally, as previously stated, in the last three influenza seasons, according to a Decree of the Regional Health Department, a mandatory dissent form was required to those HCWs who declined seasonal influenza vaccination.

### 2.2. Statistical Analysis

Working days lost and consent/dissent form responses were collected in a database using software EpiInfo 3.5.1 (Epi Info™, CDC, Atlanta, GA, USA). Data were analyzed using statistical software package STATA v14.2 (StataCorp LP, College Station, TX, USA). Distribution for age classes, mean age, and standard deviation of vaccinated HCWs was analyzed.

Vaccination coverage rates were obtained considering HCWs who received influenza vaccination during the campaign among the health-care personnel working at the UH of Palermo, including medical residents and health-care trainees. The chi-square for the trend of the vaccination coverages observed during the last twelve influenza seasons was calculated trough EpiInfo 3.5.1. Absolute and relative frequencies were calculated for categorical variables (reasons for vaccination refusal). The chi-square test was also used to evaluate the statistical significance of differences in the three parameters evaluating work absenteeism in the pre and post-intervention seasons. Mean age distribution over time wasanalyzed using an ANOVA test. The significance level chosen was *p* < 0.05.

## 3. Results

Table 2 shows the distribution of health care professionals working at the University Hospital of Palermo according to type of employment, gender, age group, adherence to influenza vaccination, and influenza vaccination acceptance during the 2018/2019 influenza season.

Higher vaccination coverages were observed among medical residents (54.9%) and health-care trainees (46.1%), which represent the categories of HCWs with the youngest age. Medical doctors showed a higher acceptance rate (33.8%) than nurses, midwives, and health-care assistants (23.9%).

Influenza vaccination coverages among HCWs of UH of Palermo from 2007/2008 to 2018/2019 seasons are shown in Figure 1. During the 2018/2019 influenza season, the vaccination coverage rate increased consistently, reaching a value of 37.2%, even higher than 25% observed during the 2009/2010 season (considering the adherence to the two trivalent and AH1N1 “pandemic” influenza vaccines available).

After the 2009/2010 season, a decreasing trend of vaccination coverage was registered until 2014/2015, with an average value of 5% during five consecutive seasons. From the beginning of the communicative and informative interventions, an increasing trend of vaccination adherence was observed from 16.7% in the 2015/2016 season to 37.2% in 2018/2019.

An overall 24.9% increase in vaccination adherence comparing 2014/2015 and 2018/2019 was reported with a significant chi-square for a trend of 550.9 (*p*-value < 0.001).

In Figure 2, the age group distribution of vaccinated HCWs during the last six influenza seasons (2013/2014–2018/2019) was reported. In particular, in the last four seasons, there was a considerable increase the proportion of vaccinates of HCWs aged from 21 to 30 years (an age group consistent with medical residents and health-care trainees), which represented 40.8% of all subjects vaccinated in 2016/2017, 29.5% in 2017/2018, and 52% in 2018/2019 seasons.

Moreover, also the percentage of HCWs aged from 31 to 40 years progressively increased over time, reaching 25% of the overall HCWs vaccinated during the 2017/2018 season. Consequently, the mean age of vaccinated HCWs had a progressive and significant decreasing trend from 48.1 (95% CI: 45.7–50.5) in the 2013/2014 season to 35.9 (95% CI: 35.0–36.8) in the 2018/2019 season (*p*-value < 0.001).

Reasons for not receiving seasonal influenza vaccination, reported in the dissent form during the last three vaccination seasons, are shown in Figure 3. In the 2016/2017 season, fear of adverse reaction (32%) followed by not considering themselves as a high-risk group for contracting influenza (24%) were the main reasons reported. In 2017/2018, both previously mentioned reasons were equally reported as principal (33%) justifications for vaccination refusal. Furthermore, in 2018/2019, not considering HCWs as a high-risk group for contracting influenza (32%) overcame fear of adverse reaction (26%) as the main reason for not vaccination. In the analysis of vaccination refusal, no significant difference emerged between the age groups and types of employees.

Finally, in Table 3, the seasonal average number of HCWs absent from work, the seasonal average number of working days lost, and the seasonal average working days lost for single HCW due to acute sickness were reported. In particular, a comparison of these data between pre (from 2009/2010 to 2014/2015) and post (from 2015/2016 to 2018/2019) influenza vaccination coverage rates increase, was carried out. In the post-intervention seasons, a decrease equal to 8.8% of seasonal HCWs absent from work (*p* < 0.05), to 12.9% of number of working days lost during influenza seasons (*p* = 0.05) and to 11.1% of working days lost for single HCW (*p* > 0.05), due to acute sickness, was observed at the UH of Palermo.

## 4. Discussion

The present study analyzed influenza vaccination adherence and absenteeism from work of HCWs working at the major teaching Hospital of Sicily. In particular, the impact of informative and communicative public health strategies on vaccination adherence and the reduction, as a consequence of increasing influenza vaccination coverage rates, of work absenteeism due to acute sickness during influenza seasons were evaluated.

Considering the trend of vaccination coverage at the UH of Palermo, from the low rates observed from the 2010/2011 to 2014/2015 seasons (average rate: 5%), a significant increase during the last four influenza seasons was demonstrated.

The age distribution of vaccinated HCWs during the last four seasons suggested, especially among young HCWs, a satisfactory effectiveness of informative and communicative campaigns based mainly on social networks and new social media, as previously reported in other experiences [20,21,22]. In particular, the significant decrease of the mean age of HCWs vaccinated confirmed a better response to strategies promoted at the UH of Palermo among health-care professionals under 40 years old.

Younger employees, such as medical residents, represent HCWs of the future, and the high vaccination coverage rates in their age classes are promising. However, low vaccination coverage reported among older HCWs should represent a public health issue because they have an increased risk of developing severe and complicated influenza infections that could lead to prolonged absence from work [23].

Vaccination coverage rates reached in the 2018/2019 season (37.2%) represent the best value obtained in the last twelve years at the UH of Palermo, overtaking adherence observed during the 2009/2010 “pandemic season”.

In particular, during the 2009/2010 influenza season, vaccination coverage of HCWs in Italy and Europe resulted from the sum of two different influenza vaccine uptake rates (pandemic AH1N1 and trivalent inactivated influenza vaccine) that were available [24].

At the same time, the remarkable collapse of vaccination adherence during the following seasons (from 2010/2011 to 2014/2015) was probably due to doubts about vaccine efficacy and safety, rising after “pandemic season” among the general population and also HCWs [25].

In addition, the occurrence of two suspected deaths within 48 h after adjuvanted trivalent influenza vaccine administration in November 2014, which brought the Italian Medicines Agency (AIFA) to withdraw the associated lots of vaccine as a precautionary measure, contributed to the general lack of confidence and adherence to influenza vaccination in Italy [26].

The organization of informative and communicative interventions on influenza vaccination represented an undisputable need for HCWs working at the UH of Palermo from the beginning of the 2015/2016 influenza season.

Every single intervention conducted during the past years in other settings was associated with modest increases in vaccination adherence [27,28].

A study conducted in the US showed that vaccination coverage among HCWs, who reported at least two interventions, was about twice than HCWs without any intervention at the workplace [29]. These results were similar to those shown by a recent review on interventions to increase influenza vaccination coverage among HCWs, which reported that vaccination uptake had a direct relation with multiple intervention strategies adopted in health care settings [30,31,32]. For all these reasons, a “multiple interventions” strategy was developed to guarantee a substantial increase in vaccination coverage rates among HCWs at the UH of Palermo.

Among HCWs, influenza vaccination coverages higher than 75% were often observed in Countries with mandatory vaccination policies such as the United States [33,34]. Similarly, in Germany, a compulsory policy requiring to wear a protective mask during the work shift was introduced for non-vaccinated HCWs in direct contact with patients, causing a consistent and fast increase of vaccination coverage rates [35].

Influenza vaccination rates raised during the last four influenza seasons at the UH of Palermo were higher than the Italian context but remained still lower than other European Countries [36,37].

Moreover, in Italy and Sicily, influenza vaccination adherence among the general population was stable during the last seasons (about 15%). Among the elderly, for which influenza vaccination was actively and offered free of charge, coverages reached 53.1% during the 2018/2019 season, showing a steady trend in comparison to previous seasons [38].

The reduction of absenteeism from work due to acute sickness of HCWs of the UH of Palermo, during influenza seasons with higher coverage rates, could confirm the effectiveness of vaccination in the reduction of influenza transmission within the health-care setting [39].

In the future, in order to effectively address immunization gaps, it is necessary to understand the main barriers for influenza vaccination adherence at different levels, planning evidence-based informative and communicative interventions for HCWs. In particular, among HCWs of the UH of Palermo, the main reasons for influenza vaccination refusal in previous years were principally focused on personal issues of HCWs (fear of adverse reaction, misperception of risk to contract influenza), rather than considering the effectiveness of influenza vaccination for patients protection [40,41]. Institutional informative and communicative strategies at the UH of Palermo contributed to an increase in vaccination coverage [42,43]. Only with an integration with tailored informative and communicative strategies and better university training, false perceptions on influenza vaccination could be modified over time among HCWs.

Following an Italian outbreak of measles, the National Health Authority tried to address the question of poor HCWs vaccination coverage approving the “Law on Prevention and Vaccination” at the end of 2017 [44].

The Law, integrated with the recent National Vaccination Plan 2017–2019 that strongly recommends vaccinations for HCWs against vaccine-preventable diseases, required a compulsory self-declaration for every health-care and socio-sanitary professional, proving their immunization status [45,46,47].

Two main limitations affected the study. The first limitation is related to the fluctuation of the HCWs population during consecutive influenza seasons that could reduce the association of interventions conducted with vaccination adherence increase. Notwithstanding, the replacement of HCWs is mainly due to a little proportion of medical residents that obtain a medical specialty degreeevery year. So, the majority of the population received all the interventions during the last four influenza seasons, suggesting a great impact on vaccination coverage rates of the strategies adopted. Moreover, to reduce the impact of health-care personnel fluctuation on work absenteeism data, a standardization of average working days lost for the number of HCWs of the UH of Palermo was carried out.

The second limitation concerns the evaluation of the absenteeism from work. Data were selected only for sick days due to acute illness affecting HCWs of the UH of Palermo during influenza seasons, for which not only influenza but also other microbial agents could be responsible. Nevertheless, influenza, among all respiratory viruses and bacteria, has the highest impact on staff absenteeism during the cold season in Europe and in Italy [48,49].

## 5. Conclusions

The present study confirmed a substantial low adherence and poor attitudes regarding influenza vaccination among older HCWs. The impact of communicative and informative strategies in order to improve influenza vaccination awareness and adherence of HCWs was more successful among younger HCWs, such as medical residents and health-care trainees. Despite the clear improvement of vaccination coverage rates and the substantial reduction of work absenteeism observed after interventions were conducted, the overall adherence still remains below that recommended by National and International Public Health Authority, suggesting the need for mandatory influenza vaccination policies for HCWs [45].

## Figures and Tables

**Figure 1 vaccines-08-00005-f001:**
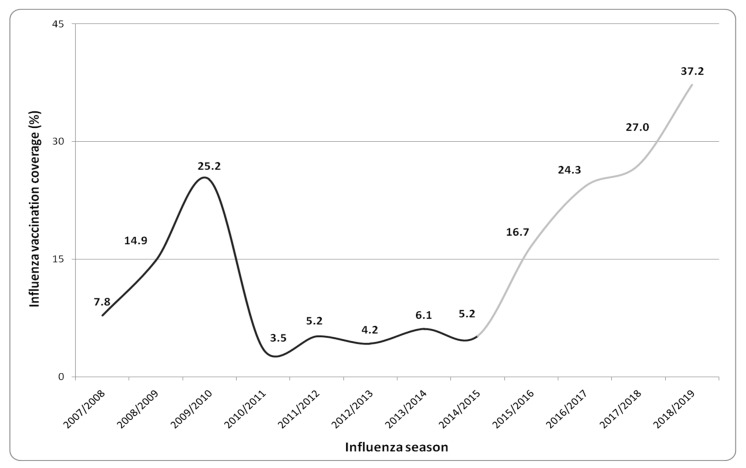
Trend of influenza vaccination coverage rates of health-care professionals working at the University Hospital of Palermo from the 2007/2008 to 2018/2019 season. (In light grey, the seasons during which the different strategies described in Table 1 were adopted).

**Figure 2 vaccines-08-00005-f002:**
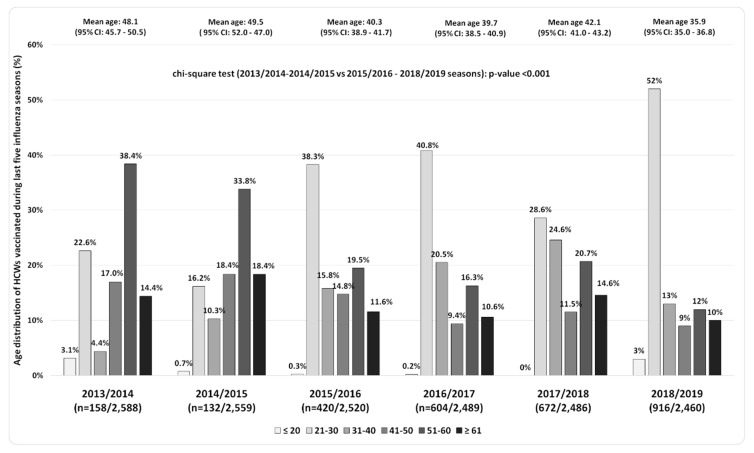
Age distribution of vaccinated HCWs during the last six influenza seasons at the University Hospital of Palermo (2013/2014–2018/2019).

**Figure 3 vaccines-08-00005-f003:**
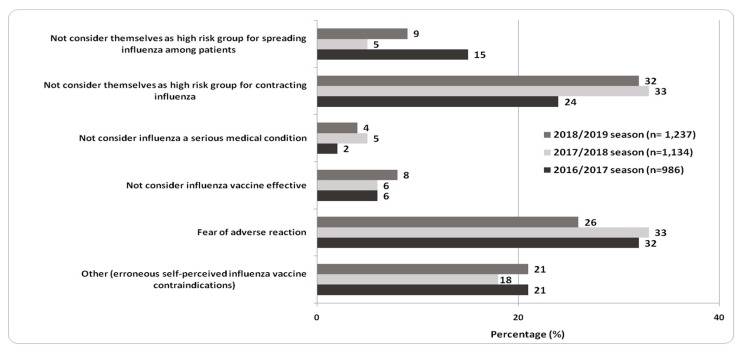
Reasons for not receiving influenza vaccination reported in the compulsory dissent form, collected from HCWs of the UH of Palermo in the influenza seasons 2016/2017, 2017/2018, and 2018/2019.

**Table 1 vaccines-08-00005-t001:** Strategies adopted by the University Hospital of Palermo to increase influenza vaccination coverages among health care workers (HCWs) in the last four influenza seasons.

Influenza Seasons	Strategies Adopted					
	Pins Distribution	“Flu Vaccination” Day	Posters and Flyers in Hospital Units	“Social Media” and Web Campaign	“On-Site” Vaccination	Mandatory Compilation of Dissent form
2015/2016	X			X	X	
2016/2017		X	X	X	X	X
2017/2018		X	X	X	X	X
2018/2019			X	X	X	X

**Table 2 vaccines-08-00005-t002:** Descriptive analysis of HCWs of the University Hospital (UH) of Palermo during the 2018/2019 influenza season.

	Medical Doctors (450; 18.3%)	Medical Residents (796; 32.3%)	Health-Care Trainees (102; 4.1%)	Nurses/Midwives/ Health-Care Assistants (768; 31.2%)	Health-Care Technicians (344; 14.1%)	Overall (*n* = 2460)
***n* (%)**						
**Gender**						
male	282 (67.2)	261 (32.8)	36 (35.3)	267 (34.8)	198 (57.6)	1044 (42.4)
female	168 (37.3)	535 (77.2)	66 (64.7)	501 (65.2)	146 (42.4)	1416 (57.6)
**Age groups**						
≤30	0 (0)	753 (94.6)	102 (100)	28 (3.6)	0 (0)	883 (35.9)
31–40	9 (2)	43 (5.4)	0 (0)	92 (12.0)	8 (2.3)	152 (6.1)
41–50	125 (27.8)	0 (0)	0 (0)	213 (27.7)	85 (24.7)	423 (17.2)
51–60	182 (40.4)	0 (0)	0 (0)	314 (40.9)	123 (35.7)	619 (25.2)
≥61	134 (29.8)	0 (0)	0 (0)	121 (15.7)	128 (37.2)	383 (15.6)
**Vaccination adherence**	152 (33.8)	437 (54.9)	47 (46.1)	184 (23.9)	86 (25.0)	916 (37.2)
**At least three influenza vaccinations during last five influenza seasons**	27 (37.5)	75 (17.2)	0 (0)	112 (60.9)	66 (76.8)	280 (30.6)

**Table 3 vaccines-08-00005-t003:** Data on absenteeism from work due to acute sickness during pre and post intervention influenza seasons among HCWs of the UH of Palermo.

Observation Period: from 1 November to 31 March	Pre-Intervention Influenza Seasons (2009/2010–2014/2015)	Post-Intervention Influenza Seasons (2015/2016–2018/2019)	% Reduction
Average seasonal number of HCWs absent from work due to acute sickness (95% CI)	1858 (1797–1919)	1693 (1573–1813)	8.8
Average seasonal number of working days lost due to acute sickness (95% CI)	11,571 (11,023–12,119)	10,077 (8626–11,528)	12.9
Average seasonal number of working days lost for single HCW due to acute sickness (95% CI)	4.5 (4.3–4.7)	4.0 (3.4–4.6)	11.1

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
