# Peer review of "Impact of Communicative and Informative Strategies on Influenza Vaccination Adherence and Absenteeism from Work of Health Care Professionals Working at the University Hospital of Palermo, Italy: A Quasi-Experimental Field Trial on Twelve Influenza Seasons"

_vaccines, 2019, doi:10.3390/vaccines8010005_

Round 1

Reviewer 1 Report

I believe the authors have appropriately addressed a majority of major concerns with the study as previously submitted. 

Author Response

Reviewer #1

Comment: I believe the authors have appropriately addressed a majority of major concerns with the study as previously submitted. 

A: Dear Reviewer, thank you for your evaluation of the manuscript. Your suggestions raised during round 1 of revision were very useful to improve our manuscript

Reviewer 2 Report

Table 2 describes the distribution of health care professional during 2018/2019 season , but in the line 149 is written "during last five influenza season". Please clarify this inequality.

Author Response

Reviewer #2

Comment: Table 2 describes the distribution of health care professional during 2018/2019 season , but in the line 149 is written "during last five influenza season". Please clarify this inequality.

A: According to the appropriate suggestion, the mistake was corrected. Table 2 reports data of health care professionals of the UH of Palermo during last season (2018/2019) examined.

Reviewer 3 Report

This manuscript studies the effect of implementing several influenza vaccine initiatives (e.g. ) on improving healthcare workers’ vaccination coverage and reducing their absenteeism. The study is well designed and scientifically sound. I only have a few comments, as follows:

(1) The mean age of vaccinated HCWs was decreased from 48.1 (2013/2014) to 35.9 (2018/2019), as shown in Figure 2. Could you briefly discuss the reason for this changing pattern?

(2) Confidence intervals are needed in Table 3. Currently, the authors only provided the mean values averaged over all studied years.

(3) The authors mainly discussed the vaccination coverage. What is the influence of vaccine efficacy across the studied years? For example, did you observe mismatch between vaccinated strains and truly circulated strains?

Author Response

Reviewer #3

Comment: This manuscript studies the effect of implementing several influenza vaccine initiatives (e.g. ) on improving healthcare workers’ vaccination coverage and reducing their absenteeism. The study is well designed and scientifically sound. I only have a few comments, as follows.

A: Dear reviewer, Thank you for your precise and accurate revision and for the opportunity to revise and improve our manuscript. We hope that this revised version could fulfil all the suggestions raised.

Comment: The mean age of vaccinated HCWs was decreased from 48.1 (2013/2014) to 35.9 (2018/2019), as shown in Figure 2. Could you briefly discuss the reason for this changing pattern?

A: According to your suggestion, we integrated in the discussion section the sentence " The age distribution of vaccinated HCWs during the last four seasons suggested, especially among young HCWs, a satisfactory effectiveness of informative and communicative campaigns based mainly on social networks and new social media, as previously reported in other experiences" with "In particular, the significant decrease of the mean age of HCWs vaccinated confirmed a better response to strategies promoted at the UH of Palermo among health-care professionals under 40 years old."

Comment: Confidence intervals are needed in Table 3. Currently, the authors only provided the mean values averaged over all studied years.

A: As correctly suggested, the confidence intervals were added in table 3.

Comment: The authors mainly discussed the vaccination coverage. What is the influence of vaccine efficacy across the studied years? For example, did you observe mismatch between vaccinated strains and truly circulated strains?

A: As correctly stated, we mainly discussed vaccination coverage trends and their modification according to strategies adopted. Unfortunately, an evaluation of vaccine effectiveness and of the possible mismatch (due to B strain mismatch before 2014/2015 when trivalent vaccines were administered or to AH3N2 strain mutation during influenza season) among HCWs was not carried out. In future, a study on vaccine effectiveness of influenza vaccination at the UH of Palermo will be conducted. Specifically, during the current influenza season (2019/2020) a virological surveillance of ILIs occurred among vaccinated and not vaccinated HCWs of the UH of Palermo will be performed.
